

# Uncertainties from biomass burning aerosols in air quality models obscure public health impacts in Southeast Asia

Margaret R. Marvin[1,a], Paul I. Palmer[1,2], Fei Yao[1,2], Mohd Talib Latif[3], and Md Firoz Kahn[4,b]

[1]National Centre for Earth Observation, University of Edinburgh, Edinburgh, UK
[2]School of GeoSciences, University of Edinburgh, Edinburgh, UK
[3]Department of Earth Sciences and Environment, Faculty of Science and Technology, Universiti Kebangsaan Malaysia, Bangi, Malaysia
[4]Department of Chemistry, Faculty of Science, Universiti Malaya, Kuala Lumpur, Malaysia
[a]now at: Air Resources Laboratory, National Oceanic and Atmospheric Administration, College Park, MD, USA and Center for Spatial Information Science and Systems, George Mason University, Fairfax, VA, USA
[b]now at: Department of Environmental Science and Management, North South University, Dhaka, Bangladesh

**Correspondence:** Margaret R. Marvin (margaret.r.marvin@noaa.gov)

**Abstract.** Models suggest that biomass burning causes thousands of premature deaths annually in Southeast Asia due to excessive exposure to particulate matter (PM) in smoke. However, measurements of surface air quality are sparse across the region, and consequently estimates for the public health impacts of seasonal biomass burning are not well constrained. We use the nested GEOS-Chem model of chemistry and transport (horizontal resolution of $0.25° \times 0.3125°$) to simulate atmospheric composition over Southeast Asia during the peak burning months of March and September in moderate burning year 2014. Model simulations with GEOS-Chem indicate that regional surface levels of $PM_{2.5}$ (fine particulate matter with a diameter of $<$ 2.5 microns) greatly exceed world health guidelines during the burning seasons, resulting in up to 10,000 premature deaths in a single month. However, the model substantially underestimates the regional aerosol burden compared to satellite observations of aerosol optical depth (AOD) (20–52%) and ground-based observations of PM (up to 54%), especially during the early burning season in March. We investigate potential uncertainties limiting the model representation of biomass burning aerosols and develop sensitivity simulations that improve model-measurement agreement in March (to within 31%) and increase the estimated number of $PM_{2.5}$-related premature deaths that month by almost half. Our modifications have a much smaller impact on the same metrics for September, but we find that this is due to canceling errors in the model. Compared to $PM_{2.5}$ simulated directly with GEOS-Chem, $PM_{2.5}$ derived from satellite AOD is less sensitive to model uncertainties and may provide a more accurate foundation for public health calculations in the short term, but continued investigation of uncertainties is still needed so that model analysis can be applied to support mitigation efforts. Further reduction of uncertainties can be achieved with the deployment of more aerosol measurements across Southeast Asia.

## 1 Introduction

Widespread biomass burning contributes to unhealthy living conditions across Southeast Asia, collectively home to a population of more than 655 million people. Distinctive climatologies between mainland (Cambodia, Laos, Myanmar, Thailand, Viet-





nam, and Peninsular Malaysia) and maritime (Brunei, Indonesia, Singapore, Timor-Leste, East Malaysia, and the Philippines) Southeast Asia result in two burning seasons every year. These burning seasons coincide with dry conditions on the mainland in November–May and the across the more equatorial maritime nations in June–October (Duncan et al., 2003; Csiszar et al., 2005). Although seasonal burning patterns are fairly consistent, large-scale climate variations (e.g., the El Niño Southern Oscillation) can drastically affect the extent and intensity of regional fire activity from year to year (van der Werf et al., 2008; Wooster et al., 2012; Marlier et al., 2013; Field et al., 2016; Huijnen et al., 2016). With such a large population at risk, it is of utmost importance to ensure that the public health impacts of biomass burning in Southeast Asia are well understood.

The combustion process intrinsic to biomass burning results in the emission of numerous gases and aerosols, some of which are hazardous to human health. Of primary concern is $PM_{2.5}$ (fine particulate matter with a diameter of < 2.5 microns), which can impede normal functioning of the heart and lungs when inhaled by humans, leading to an increased risk of premature death (Atkinson et al., 2014; Yorifuji et al., 2015). Although total $PM_{2.5}$ is complex in composition and may originate from a variety of sources, pyrogenic $PM_{2.5}$ is primarily comprised of particulate organic carbon (OC) (Wooster et al., 2018), some of which is emitted directly and some is produced by emitted gases via secondary atmospheric chemistry (Akagi et al., 2011; Yokelson et al., 2013; Stockwell et al., 2015). Current guidelines from the World Health Organization (WHO) recommend that short-term (24-hour) exposure to $PM_{2.5}$ should not exceed 15 $\mu$g m$^{-3}$ (World Health Organization, 2021). However, ground-level $PM_{2.5}$ concentrations in Southeast Asia are often much higher, especially during the burning seasons. Atmospheric chemistry models have been applied to estimate the public health impacts of elevated $PM_{2.5}$ from severe fire events in Southeast Asia (Marlier et al., 2013; Crippa et al., 2016), reporting in some cases up to 100,000 attributable deaths (Koplitz et al., 2016). Studies like these tend to focus on extreme scenarios, and less is currently known about the public health impacts of biomass burning in Southeast Asia during more typical burning years.

Models are helpful tools for simulating atmospheric composition and assessing public health, but they are by definition limited by uncertainties in the underlying knowledge, and there are few measurements in Southeast Asia available for model evaluation. Satellite observations of aerosol optical properties provide the best coverage in time and space, but models are often needed to relate remote measurements with air quality conditions on the ground (van Donkelaar et al., 2010, 2015; Boys et al., 2014; Hammer et al., 2020; Yao and Palmer, 2021). Continuous ground-based monitoring of surface $PM_{2.5}$ has been historically scant throughout Southeast Asia, even among countries with extensive air quality networks (e.g., Malaysia, Thailand, and Indonesia). In recent years, however, Malaysia in particular has made considerable progress in expanding its network to include more measurements of $PM_{2.5}$ (Ab. Rahman et al., 2022; Ahmad Mohtar et al., 2022), and its central location spanning both mainland and maritime Southeast Asia is favorable for observing air quality during both burning seasons. In situ measurements can provide further information about the emissions and composition of $PM_{2.5}$, but previous fire-focused field experiments in Southeast Asia have been limited to certain ground sites, marine campaigns, and extreme events (Lin et al., 2013; Wooster et al., 2018).

Here, we use satellite observations across Southeast Asia and ground-based measurements from Malaysia to evaluate uncertainties related to pyrogenic aerosols in the GEOS-Chem atmospheric chemistry model for moderate burning year 2014. In a previous study (Marvin et al., 2021), we characterized biomass burning emissions in 2014 and identified two distinct



regimes: (1) burning on the mainland peaking in March and (2) burning in Indonesia peaking in September. The first regime is primarily attributed to deforestation with minor contributions from the burning of peat and savanna, whereas peat becomes the dominant fuel later in the year. The type and amount of vegetation burned in each regime determines the composition of biomass burning emissions and ultimately the overall impact on regional air quality and public health. This study focuses on

$PM_{2.5}$, and the model and data used here are described in Sect. 2. In Sect. 3, we evaluate the simulated aerosol burden over Southeast Asia across March and September of 2014. In Sect. 4, we discuss uncertainties that limit the model representation of biomass burning aerosols, and then we investigate model sensitivity to certain uncertainties in Sect. 5. We report on the public health implications of unresolved aerosol uncertainties in Sect. 6 and highlight the advantages of satellite-derived $PM_{2.5}$ for public health calculations in Sect. 7. We conclude this study in Sect. 8.

## 2  Model and data


Here, we describe the GEOS-Chem model of atmospheric chemistry and transport, as well as the set of observations that we use to evaluate simulated aerosols over Southeast Asia.

### 2.1  The GEOS-Chem model

We use version 12.5.0 of the 3-D GEOS-Chem model (www.geos-chem.org, last access: 3 February 2023) to simulate atmo-

spheric composition over Southeast Asia in 2014. Following one year of model spin-up, we run the global model at a horizontal resolution of $2° \times 2.5°$ for all of 2014, from which we extract boundary conditions that we use to run the nested model over a regional domain of $-10$ to $24°$ N and 90 to $140°$ E (Fig. 1) at a finer resolution of $0.25° \times 0.3125°$ for the months of March and September. All of our simulations extend vertically through 47 terrain-following sigma levels between the surface and 0.01 hPa.

Model inputs used in this work are replicated from Marvin et al. (2021). For example, the model is driven by assimilated meteorology from the GEOS Forward Processing (GEOS-FP) product, except for the spin-up run, which uses the Modern-Era Retrospective analysis for Research and Applications version 2 (MERRA-2) due to the unavailability of GEOS-FP before 2014. Both GEOS-FP and MERRA-2 are provided by the Global Modeling and Assimilation Office (GMAO) at NASA Goddard Space Flight Center. Anthropogenic emissions are supplied on a global scale by the Community Emissions Data System

(CEDS) (Hoesly et al., 2018) but are replaced by the regional MIX inventory over Asia (Li et al., 2017). Biogenic emissions of volatile organic compounds (VOCs) are calculated online using the Model of Emissions of Gases and Aerosols from Nature (MEGAN) version 2.1 (Guenther et al., 2012), and natural emissions of nitrogen oxides ($NO_x = NO + NO_2$) are parameterized (Hudman et al., 2012; Murray et al., 2012). As in Marvin et al. (2021), we primarily use biomass burning emissions from the Global Fire Emissions Database (GFED) version 4.1s (van der Werf et al., 2017), though we also test other inventories that are

compatible with GEOS-Chem (Section 4.1). The base GFED4.1s inventory has a spatial resolution of $0.25° \times 0.25°$, and we configure GEOS-Chem to apply daily and 3-hourly scaling factors to the monthly data so that we can achieve finer temporal resolution in the nested simulation.




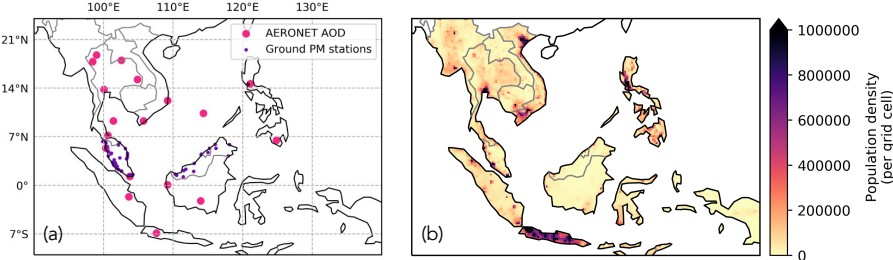

**Figure 1.** Nested model domain over Southeast Asia. Panel a) shows the locations of AERONET AOD (hot pink) and ground-based PM monitoring sites (purple) that we use in our study. Panel b) shows the regional population density (per grid cell) that we estimate for the region, based on the $0.25° \times 0.25°$ gridded population data for 2015 from NASA SEDAC and scaled to 2014 using bulk population data from the World Bank as described in Sect. 3.1.

Atmospheric chemistry in all of our simulations is described by the "complexSOA_SVPOA" GEOS-Chem mechanism, which is based on the full-chemistry "tropchem" mechanism for gas-phase reactions (Eastham et al., 2014) but also accounts for the photochemical production of secondary organic aerosols (SOA) and semi-volatile primary organic aerosols (SVPOA). A standard volatility basis set (VBS) scheme is used to estimate the yield of SOA families from their respective primary VOCs (Pye et al., 2010), and the formation of isoprene SOA is additionally represented by aqueous uptake from its immediate gas-phase precursors (Marais et al., 2016). The model generates 3-D fields of mass concentrations for organic aerosols, inorganic aerosols, sea salt aerosols, black carbon, and dust. Model $PM_{2.5}$ is calculated as the total mass concentration of those aerosols that exist in the fine mode. Type-specific hygroscopic growth factors are applied at 35% relative humidity, consistent with sampling conditions for the corresponding observations. Model aerosol optical depth (AOD) represents aerosol extinction coefficients, integrated across each vertical layer, and is reported in our simulations at 550 nm. We have updated the model calculations for AOD and $PM_{2.5}$ to account for oxidized primary organic aerosol (OPOA) and to exclude VBS-derived isoprene SOA in favor of the aqueous uptake estimate. For comparison with observations, we sample model output at the time and location of the measurements described below.

## 2.2 MODIS and AERONET AOD

We use satellite and ground-based observations of aerosols to evaluate the model simulations. Space-borne observations of the total columnar AOD are obtained from the NASA Moderate Resolution Imaging Spectroradiometer (MODIS) instrument aboard the Aqua satellite (MYD04_L2), which has a local equatorial overpass time of 13:30. Although MODIS AOD is also available at an overpass time of 10:30 from the Terra satellite (MOD04_L2), we do not find a significant difference in AOD at the two overpass times during the burning season in Southeast Asia, according to a two-sample t-test (March: $p$-value



= 0.57; September: $p$-value = 0.21), and choose to focus our study instead on just the afternoon dataset which coincides with potentially useful satellite observations of aerosol precursors and other related species (Levelt et al., 2006; Veefkind et al., 2012). In particular, we use the MODIS Collection 6.1 Level 2 combined Dark Target and Deep Blue AOD product (AOD_550_Dark_Target_Deep_Blue_Combined) at 550 nm (Levy et al., 2015). This product is generated using a fixed thresholding method based on the Normalized Difference Vegetation Index (NDVI) and, on a global scale, has a Pearson correlation coefficient of 0.91, a mean absolute error of 0.067, and a root-mean-square error of 0.11 as compared to the ground-truth AERONET AOD (described below) over land for the period from 2013 to 2017 (Wei et al., 2019). While studies suggest that it is not always appropriate for the merging procedure only depending on the fixed thresholds of NDVI (Wei et al., 2019), we use this product to take advantage of its enhanced coverage in the absence of its improved versions. AOD retrievals of varying quality assurance (QA) flags coexist in the product, and we choose to use those of the best quality (AOD_550_Dark_Target_Deep_Blue_Combined_QA_Flag = 3). The retrievals are provided on a spatial resolution of 10 km, and we regrid them onto the coarser $0.25° \times 0.3125°$ model grid to facilitate a consistent comparison with our GEOS-Chem model simulations.

Ground-based observations of the total columnar AOD are obtained from the NASA AErosol RObotic NETwork (AERONET). In particular, we use the AERONET Version 3 Level 2 AOD data product that has undergone cloud screening and quality assurance (Giles et al., 2019). The AOD observations are reported at wavelengths ranging from 340 nm to 1640 nm on a high temporal frequency of up to 15 minutes. The estimated uncertainty in computed AOD, due primarily to calibration uncertainty, is ∼0.010–0.021 for field instruments, and is spectrally dependent with the higher errors in the UV (Eck et al., 1999). For each data record, we build a quadratic fit of ln(AOD) and ln(wavelength) that requires at least three valid data pairs encircling 550 nm (Eck et al., 1999), and we subsequently use the fitted relationship to interpolate AERONET AOD to 550 nm. The wavelength of 550 nm corresponds to particle sizes of 0.1–2 $\mu$m and is comparable to the $PM_{2.5}$ size range (Kahn et al., 1998). For consistency with satellite observations of AOD, we utilize AERONET data collected within ± 30 minutes of the 13:30 Aqua MODIS overpass time from 18 ground stations operating across Southeast Asia in 2014 (Fig. 1a).

## 2.3 Ground-based PM mass concentrations

Measurements of surface PM mass concentrations are provided by the Air Quality Division, Department of Environment, Malaysia. These measurements are collected at ground stations across Malaysia as part of a wider pollution monitoring network, as described by Latif et al. (2014). We use data from 59 ground stations in this network that collected measurements of PM in 2014 (Fig. 1a). At that time, Beta Attenuation Monitors (BAM) (Model 1020; Met One Instruments Inc., USA) were used to measure aerosols but reported only on $PM_{10}$ (particulate matter $\leq$ 10 microns), with no direct measurements provided for the finer subset of $PM_{2.5}$. More recently, the BAM instruments have been replaced with Tapered Element Oscillating Microbalances (TEOM) (1405-DF FDMS; ThermoFisher Scientific Inc., USA) that measure and report on both PM size ranges (Ab. Rahman et al., 2022; Ahmad Mohtar et al., 2022). For hourly data, the BAM instruments have an accuracy of within ±10%, precision within ± 5 $\mu$g m$^{-3}$, and a lower detection limit of 4.8 $\mu$g m$^{-3}$. The TEOM instruments have an accuracy of within ±0.75%, precision within ± 1.5 $\mu$g m$^{-3}$, and a lower detection limit of 0.06 $\mu$g m$^{-3}$. Using TEOM measurements





from 2018, we derive site-specific $PM_{2.5}$:$PM_{10}$ ratios, which we apply to the BAM measurements of $PM_{10}$ to infer monthly mean concentrations of $PM_{2.5}$ for 2014. Where TEOM measurements are not available (seven ground stations), we assume a ratio of 2:3 based on the mean from the remaining active locations.

## 3 Aerosol burden over Southeast Asia during the burning seasons

Here we report the regional model distribution of $PM_{2.5}$ and evaluate values against ground-based and satellite remote sensing data.

### 3.1 Regional $PM_{2.5}$ distribution

Standard simulations with the GEOS-Chem model suggest that ground-level mass concentrations of $PM_{2.5}$ greatly exceed world health guidelines across Southeast Asia during its burning seasons. Figure 2a–b shows monthly mean surface $PM_{2.5}$

mass concentrations from the nested GEOS-Chem model for March and September of 2014. During both months, mean mass concentrations often exceed 15 $\mu$g m$^{-3}$, the current 24-hour WHO limit, especially over and immediately downwind of burned areas (Fig. 2e–f), where peak values approach 100 $\mu$g m$^{-3}$. Prolonged exposure to such high levels of $PM_{2.5}$ puts regional populations at increased risk of ill health effects, even during a moderate burning year such as 2014. Assuming that each increment of 10 $\mu$g m$^{-3}$ $PM_{2.5}$ is associated with a 1.04% increase in mortality (Atkinson et al., 2014), we

apply this rate to the total number of deaths expected per month for the population of Southeast Asia as given by the gridded UN-adjusted population count product v4.11 for 2015 from the NASA Socioeconomic Data and Applications Center (https://sedac.ciesin.columbia.edu, last access: 3 February 2023), which is scaled to 2014 using national demographic data from the World Bank (https://data.worldbank.org, last access: 3 February 2023; shown in Fig. 1b). Based on the surface values for $PM_{2.5}$ simulated with GEOS-Chem, we calculate that excessive exposure to $PM_{2.5}$ is responsible for nearly 10,000

premature deaths across Southeast Asia in March of 2014 and another 7,000 in September.

### 3.2 Model evaluation

Evaluation against observations suggests that the simulated aerosol burden over Southeast Asia is underestimated by GEOS-Chem, particularly during the burning seasons. Figure 3 compares the nested model to aerosol measurements across Southeast Asia for March and September of 2014. We find that the control run underestimates monthly mean AOD across the region, with

a normalized mean bias (NMB), calculated as the mean difference between the model and observations normalized by the mean of the observations, ranging between $-20\%$ and $-52\%$. The edges of this range are defined by the comparison to AERONET AOD, whereas the comparison to MODIS AOD varies slightly less ($-31\%$ to $-39\%$), possibly smoothed by the high data density of the satellite observations ($n = 4829$ over land). In both cases, however, model-measurement agreement is markedly worse in March than September. These trends are generally supported by linear regression analysis, except that the Pearson

correlation coefficient ($r$) is weakest for MODIS AOD and in September ($r = 0.68$). Similar results are also found to describe ground-based $PM_{2.5}$ in Malaysia, which is underestimated substantially by the model in March (NMB = $-54\%$ and slope =

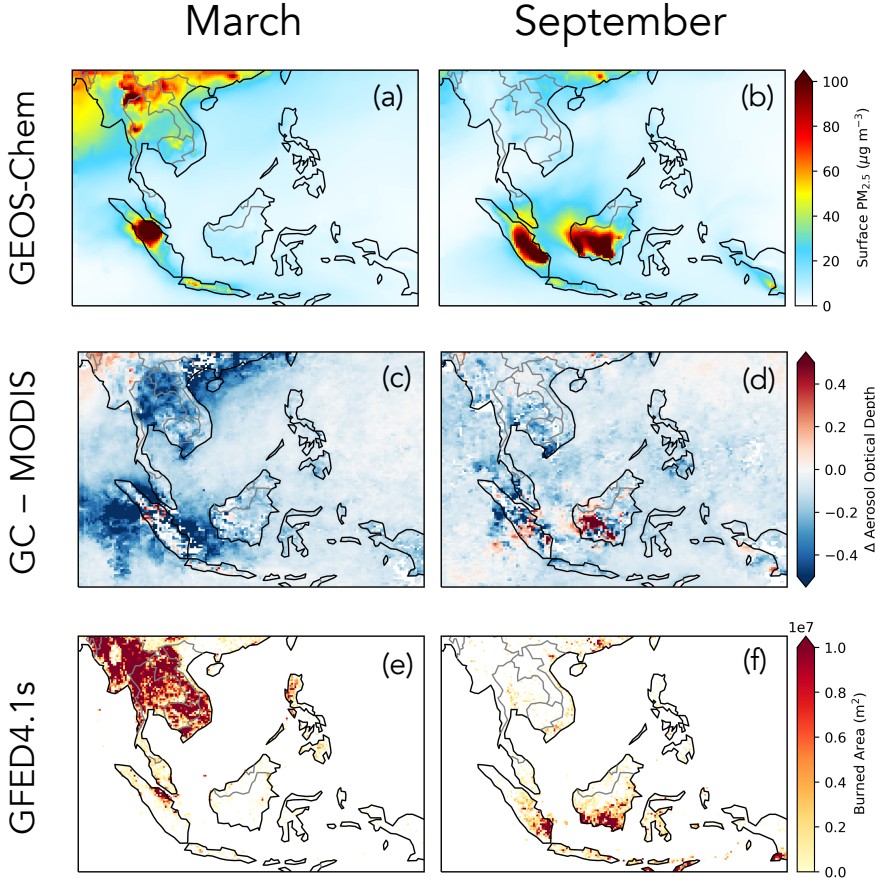

**Figure 2.** Maps of monthly mean surface $PM_{2.5}$ ($\mu g\ m^{-3}$) from the nested GEOS-Chem simulation (a–b), monthly mean difference between the nested GEOS-Chem simulation and MODIS AOD on the $0.25° \times 0.3125°$ nested model grid (c–d), and monthly burned area ($m^2$) from GFED4.1s at its native resolution of $0.25° \times 0.25°$ (e–f) over Southeast Asia in March (a,c,e) and September (b,d,f) of 2014.

0.61), and although it approaches observed values in September (NMB = 4% and slope = 1.07), $r$ is notably worse (0.68 versus 0.89). Such poor agreement with observations suggests that there are significant deficiencies that must be addressed before proceeding with further model analysis. Figure 2c–d shows that the largest deviations tend to occur over areas of fire activity, 175 and we find that the total regional model bias is minimized during the off season (e.g., for MODIS AOD in December: NMB = −10%), suggesting that biomass burning is a major source of uncertainty in simulating aerosols over Southeast Asia. We investigate this uncertainty below to better understand its impact on simulated air quality and related mortality throughout the region.

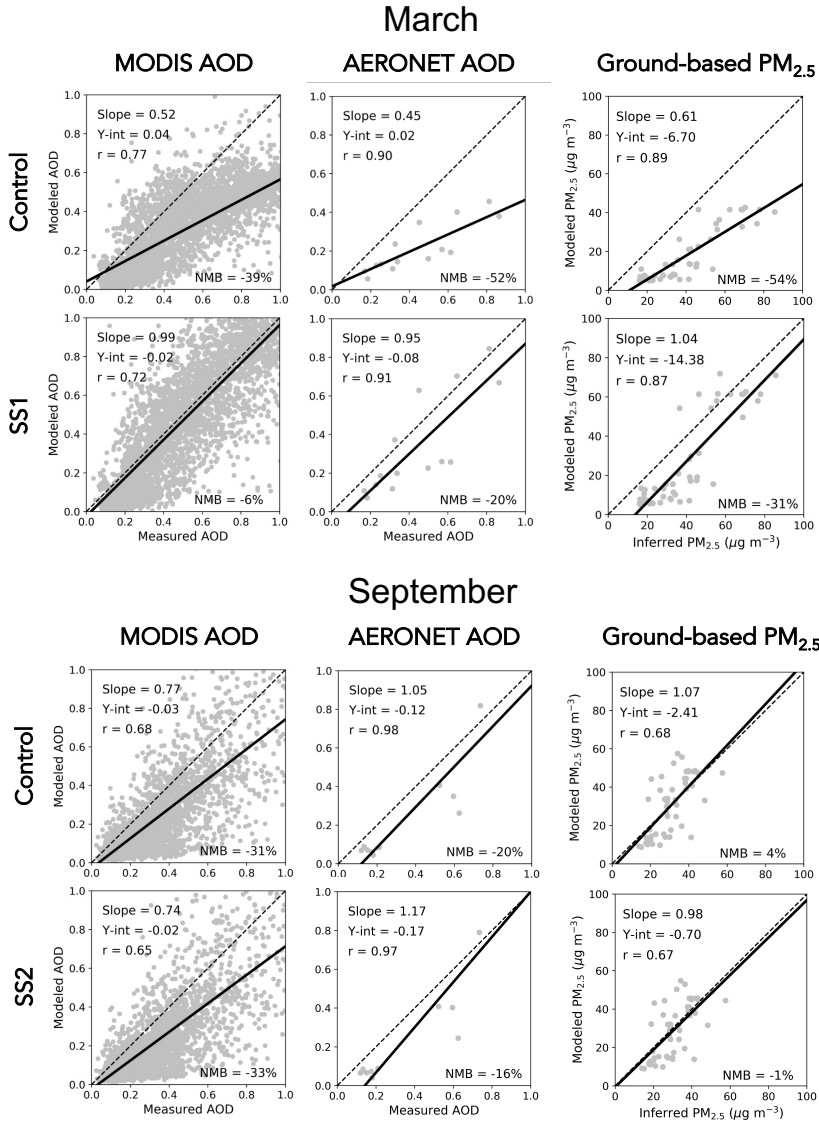

**Figure 3.** Scatterplots of modeled versus measured monthly mean MODIS AOD (over land), AERONET AOD, and surface PM$_{2.5}$ ($\mu$g m$^{-3}$) for the nested GEOS-Chem model over Southeast Asia in March and September of 2014. For each month, results are shown from the control model run as well as from the respective sensitivity simulation SS1 or SS2 as described in Sect. 5. Surface PM$_{2.5}$ concentrations are inferred from ground-based measurements of PM$_{10}$ from Malaysia as described in Sect. 2.3. Each plot shows a line of best fit (solid) and associated mean statistics (inset): the slope and $y$-intercept of the best-fit line, Pearson correlation coefficient ($r$), and normalized mean bias (NMB) as defined in the main text. The 1:1 line (dashed) is also shown for reference.





## 4 Uncertainties of biomass burning aerosols

Here we examine the impact of uncertainties associated with the emission and subsequent atmospheric transformations of biomass burning aerosols.

### 4.1 Biomass burning emissions

Uncertainties in simulating emissions of aerosols from biomass burning stem from the choice of inventory used, as well as inventory-specific burned area and fuel consumption estimates, emission factors, and injection heights.

The choice of inventory can have a significant impact on model aerosols, especially during the burning seasons in Southeast Asia (Liu et al., 2020). The default GEOS-Chem model supports four global inventories of biomass burning emissions: the Global Fire Emissions Database (GFED) (van der Werf et al., 2017), the Fire INventory from NCAR (FINN) (Wiedinmyer et al., 2011), the Global Fire Assimilation System (GFAS) (Kaiser et al., 2012), and the Quick Fire Emissions Dataset (QFED) (Darmenov and da Silva, 2015). The GFED and FINN inventories directly apply MODIS fire data to estimate biomass burning

emissions, whereas GFAS and QFED are based on MODIS-derived fire radiative power. To test variability between these inventories, we conduct simulations implementing currently supported versions GFED4.1s, FINN1.5, GFAS, and QFED2.5r1 separately into the global GEOS-Chem model. Figure 4 shows scatter plots of model versus measured AOD at the mean afternoon MODIS overpass time on the base $2° \times 2.5°$ model grid across Southeast Asia for March and September of 2014. We find that all four inventories are associated with substantial model underestimation of AOD during both months. The NMB

of the model ranges between about $-30$ and $-60\%$. We have chosen to use GFED4.1s as the basis for the remainder of our study because it consistently results in the best linear correlation between model and measured AOD (as defined by $r$), and although it does not differ much from the other inventories in March, it leads to much better agreement with measured AOD in September (slope = 0.94; NMB = $-36\%$) when peat becomes a dominant fuel for biomass burning.

### 4.1.1 Estimates for burned area and fuel consumption

The GFED4.1s inventory estimates total dry matter emissions as the product of burned area and fuel consumption (van der Werf et al., 2017). Figure 2e–f shows burned area from GFED4.1s for Southeast Asia in March and September of 2014. For the MODIS era (2000–present), burned area is supplied by the MODIS Collection 5.1 MCD64A1 product (Giglio et al., 2013), which is also combined with MODIS active fire detections (MCD14ML) to derive additional contributions from small fires ($<$ 21 ha or 500 m$^2$) (Randerson et al., 2012). Small fires are very important in Southeast Asia, as they are thought to account

for about 25–50% of the total regional burned area annually, but this estimate is highly uncertain (van der Werf et al., 2017). Furthermore, MODIS active fire products were recently updated to Collection 6 (Giglio et al., 2018), and in a subsequent study Vetrita et al. (2021) found that, compared to Collection 6, the Collection 5.1 MCD64A1 product overestimated burned area by about 35% over peatlands on Borneo in September of 2014. For that month, they found good agreement (within about 15%) between Collection 6 MCD64A1 and another burned area product called FireCCI, which is derived from Collection

6 active fire detections (MCD14ML) and like GFED4.1s is also sensitive to small fires (Lizundia-Loiola et al., 2020). To




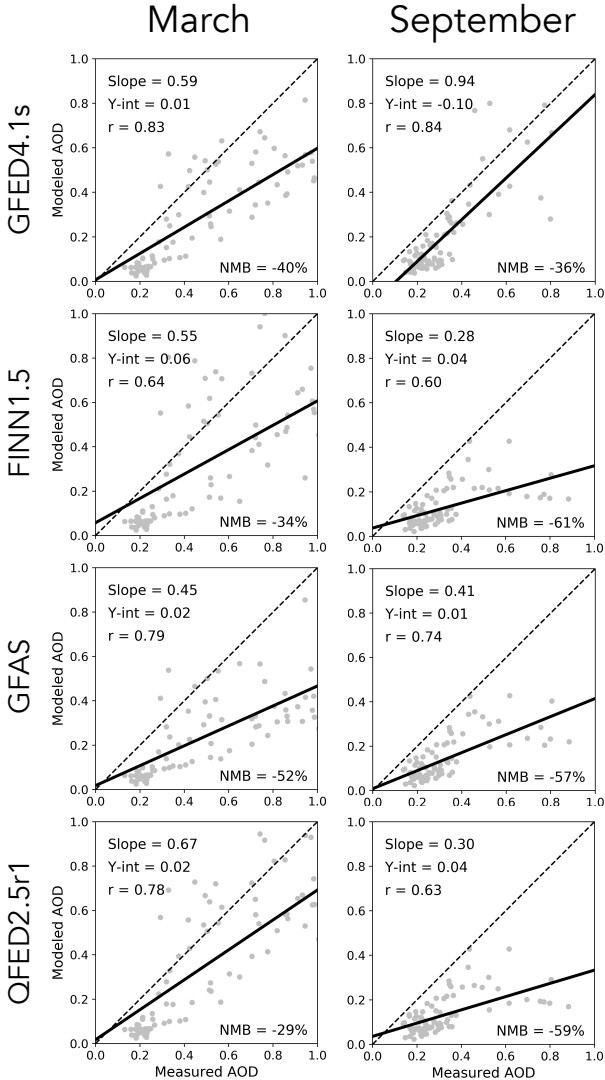

**Figure 4.** Scatterplots of modeled versus measured monthly mean MODIS AOD (over land) across Southeast Asia for March and September of 2014. For each month, model AOD is computed in GEOS-Chem on the base $2° \times 2.5°$ grid using four different biomass burning inventories: GFED4.1s, FINN1.5, GFAS, and QFED2.5r1. Each plot shows a line of best fit (solid) and associated mean statistics (inset): the slope and $y$-intercept of the best-fit line, Pearson correlation ($r$), and normalized mean bias (NMB) as defined in the main text. The 1:1 line (dashed) is also shown for reference.

evaluate uncertainties in GFED4.1s burned area, we compare to FireCCI across all of Southeast Asia in 2014. We find that the total regional burned area estimated by GFED4.1s in March ($5.2 \times 10^{10}$ m$^2$) agrees within 16% of FireCCI ($4.5 \times 10^{10}$ m$^2$). In September, however, GFED4.1s ($9.9 \times 10^9$ m$^2$) estimates nearly a factor of two more burned area than FireCCI ($5.6 \times 10^9$ m$^2$),





consistent with the high bias reported by Vetrita et al. (2021). Based on this analysis, we consider a factor of two as an upper
limit on uncertainty in GFED4.1s burned area for Southeast Asia during our study period.

Fuel consumption in GFED4.1s is parameterized based on output from the GFED modeling system and optimized using
ensemble measurements from over 120 unique locations around the world (Scholes et al., 2011; van Leeuwen et al., 2014).
Indonesia is relatively well represented in this procedure, with fuel consumption measurements assimilated from four studies at
three different locations between the islands of Sumatra and Borneo. However, no measurements are included from anywhere
else in Southeast Asia. Because relevant measurements in this region still remain scarce today, we do not currently pursue
further evaluation, but we note here that fuel consumption may be a significant source of uncertainty in GFED4.1s biomass
burning emissions, especially over mainland Southeast Asia.

### 4.1.2 Emission factors

The final stage of preparing biomass burning emissions is completed in GEOS-Chem, where global emission factors (EFs)
primarily from Akagi et al. (2011) are applied to convert the total dry matter emissions from GFED4.1s into speciated emissions
of gases and aerosols (in units of g emitted per kg dry matter burned). Recent studies have revealed significant discrepancies
between aerosol EFs measured in Southeast Asia and the global EFs applied in GFED4.1s/GEOS-Chem. For example, aircraft
measurements from Wooster et al. (2018) collected over Indonesia in 2015 indicate that the EF for $PM_{2.5}$ from peat burning is
$22.25 \pm 8.63$ g kg$^{-1}$ (with CO as reference species), of which 99% is attributed to OC. However, the EF for OC in the model is
only 6.02 g kg$^{-1}$, about 3–4 times too low. Laboratory analysis of plume samples from the same fire season produced mean EFs
for OC in the range of 12–16 g kg$^{-1}$ (Stockwell et al., 2016; Jayarathne et al., 2018), which are somewhat more conservative
than the aircraft measurements but still notably higher than the model. In a comprehensive study of the 2015 Indonesian peat
fires, Kiely et al. (2019) developed an improved estimate for particulate emissions using an average of observed EFs, resulting
in a value of 22.3 g kg$^{-1}$, similar to Wooster et al. (2018).

We attribute negative model bias to the omission of an important source of organic aerosol: the model EF, in the default setup
for GFED4.1s/GEOS-Chem, represents particulate OC but does not account for gas-phase VOCs that are of intermediate or
semi-volatility (I/SVOCs) and may subsequently partition into aerosols. Based on laboratory studies, these I/SVOCs are thought
to comprise 35–64% of the mass of non-methane organic compounds (NMOCs) emitted from biomass burning (Yokelson et al.,
2013), with EFs roughly one third of the total value for all NMOCs that have yet been identified across several different types
of vegetation burned (Stockwell et al., 2015). Although Akagi et al. (2011) do not explicitly provide EFs for I/SVOCs, they do
provide the total EFs for identified NMOC, which we scale by 1/3 for consistency with Stockwell et al. (2015). Then, we add
these values to the existing EFs for particulate OC to calculate the EFs for total OC as follows:

$$\mathrm{EF_{OC}} = \mathrm{EF_{pOC}} + \frac{1}{3}\mathrm{EF_{identifiedNMOC}}. \tag{1}$$

By accounting for the emission of I/SVOCs, we find that for peat our estimate of EF$_{OC}$ is 22.25 g kg$^{-1}$, which agrees very
well with Wooster et al. (2018) and Kiely et al. (2019). Although the limited availability of relevant measurements currently
precludes further evaluation, this correction reflects a systematic problem with the model representation of OC that we expect





applies to all vegetation types from GFED4.1s/GEOS-Chem. We note, however, that the global EFs we use here are inherently limited by measurement uncertainties, knowledge gaps, and inconsistencies between laboratory and field data (Akagi et al., 2011). More measurements are needed to evaluate existing EFs and develop improved values that better describe aerosol
emissions from biomass burning in Southeast Asia.

### 4.1.3 Vertical emissions distribution

In recent versions of GEOS-Chem including 12.5.0, the default treatment of biomass burning emissions from GFED is to inject all emissions into the surface layer of the model. This is very likely not an accurate representation of all fires in Southeast Asia, but may be appropriate in particular for peat fires, which occur close to the ground and tend to produce plumes confined to
altitudes below 1000 m (Tosca et al., 2011). In any case, we do not expect a change in the vertical distribution of biomass burning emissions to have a significant impact on total column aerosol extinction as represented by AOD. Furthermore, partial injection of emissions above the surface layer would likely result in worse agreement with ground-based $PM_{2.5}$ observations (Fig. 3). Therefore, we do not investigate alternative vertical distribution scenarios here, but we do acknowledge that such assumptions constitute yet another source of model uncertainty.

### 4.2 Organic aerosol chemistry

Once emitted, organic aerosols and their precursors may undergo secondary chemistry that affects the composition and atmospheric loading of OC. In the GEOS-Chem "complexSOA_SVPOA" mechanism, biomass burning emissions of OC are attributed to gas-phase semivolatile precursors of POA. These species may then undergo oxidation by OH to form OPOA, with a rate constant ($k_{OH}$) of $2.0 \times 10^{-11}$ cm$^3$ molec.$^{-1}$ s$^{-1}$, and the overall rate of this reaction controls the ratio of POA to
OPOA. In contrast, the contribution of SOA is derived from the emission of primary VOCs such as terpenes and aromatics, and neither OPOA nor SOA undergo further oxidation. Budisulistiorini et al. (2018) measured individual OA components at a surface site in Singapore during the Indonesian burning season of 2015 and found that the fraction of POA during that time was 40–50%. We find, however, that the model POA fraction in September of 2014 over Singapore is much lower at only 9%. This suggests that POA reactivity is too high in the model for at least one site during the late burning season, and we find much better
agreement with Budisulistiorini et al. (2018) when we reduce $k_{OH}$ by one order of magnitude (POA fraction over Singapore becomes ~40%). Figure 5 shows the regional mean distribution of the model surface OA mass concentration between its components POA, OPOA, and SOA across Southeast Asia in September of 2014. In the control run, this distribution is dominated by OPOA and SOA, with a minor contribution from POA (12%). When we apply an order of magnitude reduction to $k_{OH}$, the regional POA fraction increases to 20% and, due to the shorter lifetime of POA, results in an overall 22% reduction in the total
surface OA mass concentration. Because we are limited by the availability of other relevant measurements, we have derived this adjustment with the assumption that the OA distribution is constant between fire seasons from different years and for all burning sites across the region. Although we do not expect significant variations in POA reactivity for a given burning regime, more measurements of aerosol composition are needed to evaluate this metric across the region for both burning seasons and to explore other complicating factors such as the prevalence of SOA under different conditions in Southeast Asia.



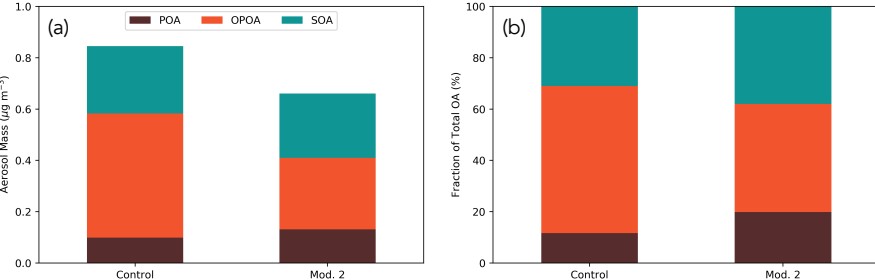

**Figure 5.** Mean model surface OA distribution in terms of a) absolute aerosol mass ($\mu$g m$^{-3}$) and b) fraction of total OA (%), separated into components POA, OPOA, and SOA across Southeast Asia in September of 2014. The left bar shows results from the control run and the right bar shows the adjusted distribution when the rate constant k$_{OH}$ controlling the oxidation of POA to OPOA is reduced by one order of magnitude (Mod. 2).

Gas-aerosol partitioning is often explicitly represented in models to describe the formation of SOA, but previous perturbation experiments have been unsuccessful in tuning related parameters to match satellite AOD and in situ OA observations simultaneously over Southeast Asia (Trivitayanurak et al., 2012). Furthermore, the SOA formation mechanism in GEOS-Chem was updated substantially in recent years (Marais et al., 2016), but multiple studies have shown that this representation of SOA does not vary significantly from more simple model configurations (Schroder et al., 2018; Jo et al., 2019; Pai et al., 2020). This

suggests that the bulk model aerosol abundances are not very sensitive to the partitioning details and that it is not likely to be a major source of uncertainty in our simulations.

### 4.3 Physical aerosol processes

Aerosol abundance is also affected by physical aerosol processes such as water solubility, hygroscopic growth, deposition, transport, and lifetime. The GEOS-Chem model assumes that POA is neither water-soluble nor hygroscopic, which is largely

consistent with recent laboratory studies of total emitted aerosol from the burning of Southeast Asian vegetation, especially peat (Chen et al., 2017, 2019; Chow et al., 2019). In contrast, OPOA and SOA are both water-soluble and hygroscopic, meaning that they are subject to water uptake from clouds and precipitation. These aerosols are lost mainly to wet deposition, which controls their lifetimes over Southeast Asia (Trivitayanurak et al., 2012). However, evaluation of recent model improvements suggests that total OC is robust against uncertainties in the GEOS-Chem wet deposition scheme globally (Luo et al., 2020). We

therefore consider the model treatment of these physical processes generally appropriate for biomass burning aerosols from Southeast Asia.

### 5 Sensitivity simulations

Considering the uncertainties of biomass burning aerosols described above, we have applied corresponding modifications to the nested GEOS-Chem model simulations for Southeast Asia in March and September of 2014 (Table 1). Sensitivity simulations



**Table 1.** Modifications as applied in sensitivity simulations of the nested GEOS-Chem model for Southeast Asia in March (SS1) and September (SS2) of 2014.

| Mod. | Description | SS1 | SS2 | Reference |
|------|-------------|-----|-----|-----------|
| 1 | Account for $EF_{SVOC}$ according to Eq. (1) | x | x | Akagi et al. (2011); Stockwell et al. (2015) |
| 2 | Reduce $k_{OH}$ of POA by one order of magnitude | – | x | Budisulistiorini et al. (2018) |
| 3 | Reduce dry matter emissions by 50% | – | x | Lizundia-Loiola et al. (2020) |

for both March (SS1) and September (SS2) include the adjustment from Eq. (1) to account for pyrogenic emission of I/SVOCs
(Mod. 1). The September simulation (SS2) additionally includes a one order of magnitude reduction in the $k_{OH}$ of POA (Mod.
2) and a 50% reduction in dry matter emissions (Mod. 3) compared to the control model run.

  We find that, compared to the control run, sensitivity simulation SS1 significantly improves model agreement with PM
observations across Southeast Asia in March. Figure 3 shows that the normalized mean bias of GEOS-Chem AOD in SS1 is
improved by about 30 percentage points against monthly mean observations from both MODIS (NMB = −6%) and AERONET
(NMB = −20%). Furthermore, the slopes from the linear regression of GEOS-Chem AOD against MODIS (slope = 0.99) and
AERONET (slope = 0.95) are increased by about a factor of two and now approach unity. The response in model $PM_{2.5}$ is
slightly weaker, as the slope of the regression against ground-based observations increases to 1.04, but the NMB only improves
by 23 percentage points and remains quite low at −31%. For all three datasets, the upward shift of scatter points is greatest
at high PM loadings, suggesting that the modifications applied in SS1 have the largest impact on conditions where PM over
land is already elevated, for example by local biomass burning activity. Low PM mass concentrations and $r$ values do not
appear to be sensitive to the same modifications, which may indicate remaining model deficiencies in environments where PM
is dominated by other sources. Although model $PM_{2.5}$ at many sites was generally below the WHO limit of 15 $\mu$g m$^{-3}$, the
observed values were higher (min = 16 $\mu$g m$^{-3}$), suggesting that while biomass burning may account for the highest PM values
in Southeast Asia, investigation of other sources may also be needed to achieve healthy living conditions at certain locations.

  In contrast, model PM does not deviate much from the control run with respect to the modifications applied to SS2 in
September. When each modification is applied separately (not shown), we find a strong increase in model PM values with
Mod. 1 (slope versus MODIS AOD = 2.04) and opposing reductions with Mod. 2 and Mod. 3. When applied together, however,
the balancing effects of these modifications result in relatively small net changes to model PM and how well it agrees with
regional observations. Figure 3 shows, for example, that the maximum change in normalized mean bias between the control run
and SS2 is about 5 percentage points, improving slightly against AERONET AOD (NMB = −16%) and ground-based $PM_{2.5}$
(NMB = −1%). This suggests that, although the control run appears to perform quite well in September, it may in fact be due
to canceling errors, resulting in the right values for the wrong reasons. We find that with SS2, model-measurement agreement
for surface PM remains very good, but there are discrepancies in regional AOD that remain to be elucidated. As with SS1, the
upward shift of AERONET AOD scatter points at high PM values suggests that our fire-based modifications have the biggest





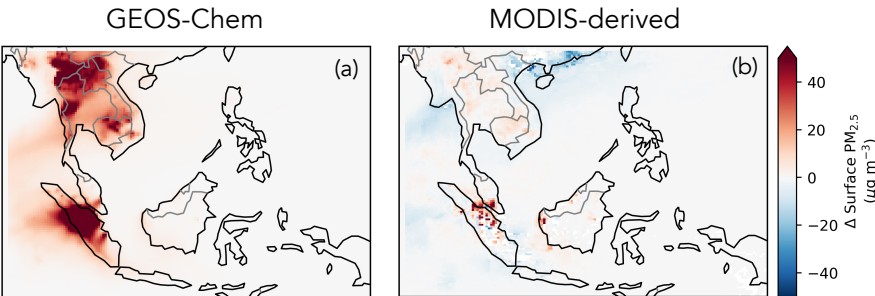

**Figure 6.** Impact of modifications from sensitivity simulation SS1 on surface PM$_{2.5}$ ($\mu$g m$^{-3}$) across Southeast Asia for March of 2014 where PM$_{2.5}$ is a) simulated directly with GEOS-Chem and b) derived from MODIS AOD.

impact near biomass burning sources, leaving less polluted environments relatively undisturbed. Considering also that burned area is smaller in September than in March (Fig. 2e–f), the statistics related to MODIS AOD (e.g., NMB = −33%) are likely to be more representative of non-pyrogenic, low-PM conditions as well. Therefore, further investigation of other sources may be required to achieve meaningful improvements to model AOD in Southeast Asia where fire activity is low.

**6  Implications for public health**

As discussed in Section 5, our sensitivity simulations indicate that uncertainties in biomass burning aerosols have the largest impact on regional aerosol loading over Southeast Asia in 2014 during the early burning season. Figure 6a shows the difference in monthly mean surface PM$_{2.5}$ between the improved simulation SS1 and the control run across Southeast Asia for March of 2014. We find that the modifications implemented in SS1 significantly increase PM$_{2.5}$ across Southeast Asia. Peak values

exceeding 40 $\mu$g m$^{-3}$ overlap with burned area and are sustained over wide expanses of land downwind. More moderate enhancements occur elsewhere, with a regional mean increase and 1$\sigma$ standard deviation of 18 $\pm$ 54 $\mu$g m$^{-3}$ over land. This difference exceeds the current 24-hour mean WHO limit of 15 $\mu$g m$^{-3}$, indicating that uncertainties in biomass burning aerosols alone can determine whether air quality conditions in Southeast Asia meet global public health guidelines.

Differences in the simulation of surface PM$_{2.5}$ also affect derived public health statistics, including the number of premature

deaths due to PM$_{2.5}$ exposure. Following the procedure described in Section 3.1, we estimate the difference in this metric between the control run and simulation SS1 for Southeast Asia in March of 2014. We find that the applied modifications lead to 4,500 additional premature deaths across the region, nearly one third of the 14,500 total deaths due to PM$_{2.5}$ that are estimated for SS1. Although we do not expect such drastic discrepancies for September of 2014, dry matter emissions (Mod. 3) vary substantially on an annual basis (van der Werf et al., 2017), and it is possible that related uncertainties may cause larger

deviations in derived public health statistics during the late burning season in other years. Continued efforts to resolve aerosol uncertainties are therefore needed across both burning seasons to ensure that model improvements remain robust over time.



## 7 Advantages of satellite-derived PM$_{2.5}$ for public health calculations

As an alternative to direct simulation, surface PM$_{2.5}$ can also be derived from satellite AOD. A common method for translating columnar AOD into ground-level PM$_{2.5}$ is to use a chemical transport model such as GEOS-Chem to determine a simple ratio between the two quantities that varies in space and time (van Donkelaar et al., 2010, 2015; Boys et al., 2014; Hammer et al., 2020). We apply daily gridded PM$_{2.5}$:AOD ratios from GEOS-Chem to AOD from MODIS to infer surface PM$_{2.5}$ mass concentrations across Southeast Asia. In March, when the control run otherwise fails to reproduce the regional aerosol burden by direct simulation, we find that satellite-derived PM$_{2.5}$ (monthly mean = $22-23$ $\mu$g m$^{-3}$) is comparable to PM$_{2.5}$ simulated directly with SS1 (monthly mean = 20 $\mu$g m$^{-3}$), no matter which simulation is used to determine PM$_{2.5}$:AOD. Substituting SS1 for the control run has very little impact on satellite-derived PM$_{2.5}$ (Fig. 6b), resulting in an absolute difference of only 82 deaths across Southeast Asia in March of 2014, which is greatly reduced from the error of 4,500 deaths attributed to the direct simulation method (Sect. 6). We highlight that the calculated premature deaths reported here should not be overinterpreted due to the simplistic nature of the method we used to calculate them. However, the considerable difference in results between the two methods does strongly suggest that satellite-derived PM$_{2.5}$ is not as sensitive to model uncertainties as simulated PM$_{2.5}$ and is instead perhaps limited by measurement error in observed AOD, which is generally well constrained and relatively small (Wei et al., 2019). Satellite-derived PM$_{2.5}$ based on model PM$_{2.5}$:AOD ratios may therefore be preferred when accuracy is needed for public health calculations in the short term, but it must be noted that the model in that case would not be fully representative of results and should not be used for further analysis unless the underlying aerosol uncertainties have been adequately addressed. Addressing these uncertainties will also benefit satellite-derived PM$_{2.5}$ products from statistical and machine learning models by providing more accurate information on the vertical distribution of aerosols, which significantly impacts their performance (Yao and Palmer, 2021).

## 8 Conclusions

We used the GEOS-Chem model to simulate air quality over Southeast Asia during its two regional burning seasons, focusing on the peak months of March and September in moderate burning year 2014. We found that the standard model simulations indicated widespread exposure to PM$_{2.5}$ at levels exceeding the current 24-hour WHO limit of 15 $\mu$g m$^{-3}$, resulting in up to 10,000 premature deaths across the region in a single month. However, substantial underestimation of the model compared to observed AOD (20–52%) and PM (up to 54%) suggests that public health statistics may be even worse than indicated by GEOS-Chem, especially in March during the early burning season.

Through a comprehensive model analysis and synthesis of the related literature, we have investigated the potential uncertainties in simulating aerosols from biomass burning and identified several model deficiencies. The most relevant and impactful of these were the fundamental omission of emitted SVOCs affecting all burning scenarios in March and September, as well as the additional overestimation of burned area and POA oxidation related to peat burning in September. Sensitivity simulations correcting these deficiencies significantly improved model underestimation in March (6–31%), and the corresponding modifications increased the estimated number of premature deaths that month by almost half. Model-measurement agreement in



September was generally much better to begin with (e.g., $PM_{2.5}$ matched within 4%), and the cumulative effect of the modifications applied in the subsequent sensitivity simulation was nearly negligible. However, the model was in fact very sensitive to the individual modifications, suggesting that canceling errors within the model have produced results that are right for the wrong reasons.

In both cases, more measurements are needed to fully characterize biomass burning aerosols across Southeast Asia. More
in situ measurements relating to the emission and composition of biomass burning aerosols are needed to constrain the uncertainties investigated in this study; more measurements of aerosols from additional sources are needed to address the remaining model bias; and more ground-based measurements of surface $PM_{2.5}$ are needed to ensure that the corresponding model analysis and public health applications are representative of the entire region. In the meantime satellite-derived $PM_{2.5}$, which our results suggest is robust against model uncertainties, can be used to achieve accuracy in public health calculations. However,
continued efforts to reduce those uncertainties are still needed so that models like GEOS-Chem can be applied effectively to mitigate the public health effects of widespread fire activity across Southeast Asia in the future.

## 9   Code and data availability

Model code and input data are free and available from the GEOS-Chem website (http://www.geos-chem.org, last access: 3 February 2023). Version 12.5.0 can be downloaded directly from https://doi.org/10.5281/zenodo.3403111 (The International
GEOS-Chem User Community, 2019).

*Author contributions.* MRM and PIP designed experiments and wrote the paper, with input and contributions from FY. FY also provided processed MODIS AOD data. MTL and MFK provided access to surface air quality data. All co-authors assisted in revising the paper.

*Competing interests.* The authors declare that they have no conflict of interest.

*Acknowledgements.* This research has been supported by the Natural Environment Research Council through the National Centre for Earth
Observation (grant nos. NE/R016518/1 and NE/R000115/1). Margaret R. Marvin, Paul I. Palmer, and Fei Yao were supported by grant number NE/R016518/1, with further support for Margaret R. Marvin from grant number NE/R000115/1. Many thanks also to the Department of Environment, Ministry of Natural Resources, Environment and Climate Change, Malaysia, for providing surface air quality data.



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
