# Peer review of "Uncertainties from biomass burning aerosols in air quality models obscure public health impacts in Southeast Asia"

_EGUsphere, 2023_

## Author Comment (AC1)

**Response to Reviewer Comments**

We thank two reviewers for their comments on the manuscript. We have responded to all individual comments below. Reviewer comments are shown in bold and our responses in plain text.

**Reviewer #1**

**Summary**

**The authors perform a series of sensitivity tests with the GEOS-Chem chemical transport model to explore how various model processes impact model performance over Southeast Asia during two biomass burning periods in 2014. The authors primarily conclude that the addition of SVOCs to default biomass burning emission inputs notably improves model performance, especially during the March time frame. The authors also note the general dearth of chemical measurements available across Southeast Asia and the need for additional observational constraints to inform model-based assessments of public health impacts from biomass burning in the region.**

**This is a useful analysis that could inform future work on mitigating public health impacts from regional and global biomass burning. However in my view there are two lines of inquiry and discussion currently missing from the analysis that should be included to justify the conclusions drawn by the authors, especially given the relevance for future public health applications.**

**General Comments**

**1.    While the authors clearly spent substantial time and resources testing the sensitivity of modeled biomass burning PM2.5 and AOD to the emission inventory used as well as several other emissions-related assumptions, there is very little discussion in the paper about the impacts of assumptions made within the AOD calculation itself. For example, Hammer et al 2016 found that increasing the assumed absorption from biomass burning aerosol in GEOS-Chem to better match satellite observations meaningfully impacted global OH. In my view it is insufficient to draw conclusions about improved model performance based on model-satellite AOD comparisons without some exploration of the AOD assumptions themselves. If an additional sensitivity simulation or two to further explore key assumptions in the AOD calculation is not feasible, perhaps at a minimum a robust discussion of these**

**factors/potential uncertainties could be added to the text to more fully contextualize the results presented. Assumed aerosol composition (smoke vs. urban vs. biogenic vs dust), optical properties by PM2.5 component (OC, EC, sulfate, nitrate, dust), and distinguishing between cloud and smoke are a few of the key areas that would seem relevant to me when it comes to comparing modeled AOD to satellite-retrieved values.**

**Hammer, M. S., Martin, R. V., van Donkelaar, A., Buchard, V., Torres, O., Ridley, D. A., and Spurr, R. J. D.: Interpreting the ultraviolet aerosol index observed with the OMI satellite instrument to understand absorption by organic aerosols: implications for atmospheric oxidation and direct radiative effects, Atmos. Chem. Phys., 16, 2507–2523, https://doi.org/10.5194/acp-16-2507-2016, 2016.**

We agree with the reviewer that this was an oversight in our discussion. We have added a new section (4.4) to discuss uncertainties related to the calculation of AOD in the model and satellite retrievals:

"4.4 Calculation of aerosol metrics

In addition to the uncertainties described above, estimates of AOD and $PM_{2.5}$ are also affected by assumptions made in the calculation of those metrics. Throughout Sect. 4, we explore aerosol composition in the model, which not only impacts the overall aerosol burden but also determines how optical properties and growth factors are assigned in the model calculations. Our discussion is primarily focused on OC from biomass burning, but we estimate that OC accounts for only up to about 50% of fine surface aerosol mass over land during the burning seasons. Other aerosols are emitted from additional sources, and their analysis is beyond the scope of this work, but their uncertainties could impact the model estimates of AOD and $PM_{2.5}$ reported here. Furthermore, the model calculations are limited by assumptions about the optical and hygroscopic properties themselves. We run GEOS-Chem with the default values for these properties, which are interpolated from look up tables generated by a Mie algorithm (Martin et al., 2003). The default values are broadly applied across lumped aerosol species and may not be accurate, depending on aerosol composition and how well it is simulated by the model. For example, global simulations suggest that underestimation of biomass burning aerosols by GEOS-Chem could potentially be explained by the absorption of brown carbon (Hammer et al., 2016; Jo et al., 2016), which is not represented in the standard model. More measurements are needed to apply this theory across Southeast Asia (Pani et al., 2021), but it is an exciting topic for future research.

Uncertainties related to the calculation of AOD in satellite and ground-based retrieval algorithms are expected to be relatively small. AERONET AOD is generally accepted as ground truth, and MODIS Collection 5 typically compares to AERONET data within an

expected error envelope of +/- (0.05 + 15%) (Levy et al., 2010). Sources of uncertainty in the satellite retrievals include radiometric calibration, estimated surface reflectance, model-derived aerosol composition, and cloud interference (Wei et al., 2019). To minimize these issues, we use the combined Dark Target and Deep Blue retrieval product, and we select only the highest quality cloud-screened data, but we acknowledge that some uncertainty will likely persist in the satellite observations."

Hammer, M. S., Martin, R. V., van Donkelaar, A., Buchard, V., Torres, O., Ridley, D. A., and Spurr, R. J. D.: Interpreting the ultraviolet aerosol index observed with the OMI satellite instrument to understand absorption by organic aerosols: Implications for atmospheric oxidation and direct radiative effects, Atmos. Chem. Phys., 16, 2507–2523, https://doi.org/10.5194/ACP-16-2507-2016, 2016.

Jo, D. S., Park, R. J., Lee, S., Kim, S. W., and Zhang, X.: A global simulation of brown carbon: Implications for photochemistry and direct radiative effect, Atmos. Chem. Phys., 16, 3413–3432, https://doi.org/10.5194/ACP-16-3413-2016, 2016.

Levy, R. C., Remer, L. A., Kleidman, R. G., Mattoo, S., Ichoku, C., Kahn, R., and Eck, T. F.: Global evaluation of the Collection 5 MODIS dark-target aerosol products over land, Atmos. Chem. Phys., 10, 10 399–10 420, https://doi.org/10.5194/ACP-10-10399-2010, 2010.

Martin, R. V., Jacob, D. J., Yantosca, R. M., Chin, M., and Ginoux, P.: Global and regional decreases in tropospheric oxidants from photochemical effects of aerosols, J. Geophys. Res. Atmos., 108, 4097, https://doi.org/10.1029/2002JD002622, 2003.

Pani, S. K., Lin, N. H., Griffith, S. M., Chantara, S., Lee, C. T., Thepnuan, D., and Tsai, Y. I.: Brown carbon light absorption over an urban environment in northern peninsular Southeast Asia, Environ. Pollut., 276, 116735, https://doi.org/10.1016/J.ENVPOL.2021.116735, 2021.

Wei, J., Li, Z., Peng, Y., and Sun, L.: MODIS Collection 6.1 aerosol optical depth products over land and ocean: Validation and comparison, Atmos. Environ., 201, 428–440, https://doi.org/10.1016/J.ATMOSENV.2018.12.004, 2019.

**2.    Perhaps this was explained and I missed it - it seems contrary to me to first emphasize the lack of available observational constraints in Southeast Asia, but then proceed to advocate for a more chemically complex modeling approach to address smoke-related applications in the region. Wouldn't the lack of measurements regionally warrant at least an exploration of simpler methods to capture biomass burning PM2.5 compared to the more sophisticated representation of organic species suggested by the authors? Maybe the answer is simply needing to dial up emissions of primary OC to begin with to address fires/burned area missed by the satellite products? Given the nature of the conclusions drawn by the authors this strikes me**

**as another line of inquiry that should be addressed somewhere, ideally through a separate sensitivity simulation to compare a simplified scaling of primary OC with both the satellite AOD and the surface PM2.5.**

A simple scaling of primary OC emissions cannot explain the model deficiencies in both March (e.g., $PM_{2.5}$ is underestimated by 54%) and September ($PM_{2.5}$ matches within 4%), but would achieve a similar result to our modification of OC emission factors (Sect. 4.1.2) that is applied in both months. In Southeast Asia, where biomass burning is dominated by peat (default EF = 6.02 g/kg), deforestation (4.71 g/kg), and savanna (2.62 g/kg), Equation 1 produces new EFs for each vegetation type (22.25 g/kg, 13.38 g/kg, and 6.75 g/kg, respectively), resulting in an overall effect that is equivalent to a simple scaling of primary OC emissions by about a factor of 3. We have inserted a new sentence at line 245 to convey this information: "During the burning seasons in Southeast Asia, the combined impact across all vegetation types is equivalent to an increase in total emissions of biomass burning OC by about a factor of 3."

**Line-by-line**

**Line 84-85. Since you mention testing other biomass burning inventories you tested, perhaps list them quickly here for clarity/easy reference?**

We have listed the inventories here as suggested.

**Line 162. Just a suggestion given the sparsity of the AERONET data points apparent in Figure 3 - did you also look at the AERONET 1.5 data? My understanding is that especially in this region a lot of data get filtered out between 1.5 and 2 during severe smoke episodes, but the filtered smoke episodes are still evident sometimes in the separation between the v1.5 fine vs. coarse product.**

We thank the reviewer for this suggestion and have investigated AERONET v1.5 as recommended. For Southeast Asia during our study period, however, we do not find any significant difference between the v1.5 and v2.0 data. Both versions produce the same number of points and overall statistics shown in Fig. 3. With the data averaged monthly, the number of points shown in the figure correspond to the number of AERONET sites reporting data across the region in March and September of 2014.

**Reviewer #2**

**Marvin et al. investigated the sources of uncertainty for biomass burning aerosols in Southeast Asia with GEOS-Chem and tried to reduce the uncertainties by adjusting several parameters in the model. The uncertainties were further related to public health impacts in the region. The manuscript is well-written and well-motivated. However, I have some concerns about the results that I think should be addressed in the revised version.**

**Major comments:**

**1. The evaluation of model aerosols seems to be limited by the availability of PM observations in the SEA. I wonder if some additional insights could be obtained from evaluation using observations of other biomass burning-related species, e.g., carbon monoxide? Since the simulations were full-chemistry in the troposphere, there should be a handful of choices.**

We thank the reviewer for this suggestion and have extended our analysis to the model representation of other biomass-burning related species using the ground-based data from Malaysia, which includes carbon monoxide (CO), nitrogen dioxide ($NO_2$), sulfur dioxide ($SO_2$), and ozone ($O_3$). Using CO as an example, because it has the largest biomass burning EFs from GFED (e.g., 210 g/kg for peat) and a relatively long atmospheric lifetime (~ 2 months), we find that the control model underestimates observed CO during both months of our study (March: NMB = -67%; September: NMB = -37%), similar to the results shown in the manuscript for $PM_{2.5}$ (Fig. 3). This could mean that the total dry matter emissions are too low in the model during both months, which would affect both CO and $PM_{2.5}$. However, like $PM_{2.5}$, the EFs applied in GFED/GEOS-Chem are significantly lower than observed in the field, particularly related to the burning of peat (e.g., 291 g/kg from Stockwell et al., 2016). The default emission factors for $NO_x$ (1.0 g/kg) and $SO_2$ (0.4 g/kg) are considerably smaller, with both species also having strong competing local sources, and we showed in a previous study that GEOS-Chem conversely captures ground-based $O_3$ in this region reasonably well (Marvin et al., 2021). Ultimately, further analysis of additional biomass burning species carries its own additional uncertainties that currently limit potential insights into the behavior of $PM_{2.5}$, and we prefer to defer this analysis for a future study.

Marvin, M. R., Palmer, P. I., Latter, B. G., Siddans, R., Kerridge, B. J., Latif, M. T., and Khan, M. F.: Photochemical environment over Southeast Asia primed for hazardous ozone levels with influx of nitrogen oxides from seasonal biomass burning, Atmos. Chem. Phys., 21, 1917–1935, https://doi.org/10.5194/acp-21-1917-2021, 2021.

Stockwell, C. E., Jayarathne, T., Cochrane, M. A., Ryan, K. C., Putra, E. I., Saharjo, B. H., Nurhayati, A. D., Albar, I., Blake, D. R., Simpson, I. J., Stone, E. A., and Yokelson, R. J.: Field measurements of trace gases and aerosols emitted by peat fires in Central Kalimantan, Indonesia, during the 2015 El Niño, Atmos. Chem. Phys., 16, 11711–11732, https://doi.org/10.5194/acp-16-11711-2016, 2016.

**2. As mentioned in the manuscript, the GEOS-Chem nested grid simulations rely on boundary conditions from a global spin-up run. Maybe I missed this point in the paper, but were the modifications listed in Table 1 applied to the global spin-up runs or only to the nested grid runs? Namely, I wonder if the authors could clarify this or discuss the potential impact of uncertainties in biomass burning from other regions close to the boundaries via long-range transport, e.g. , the southern part of China and North Australia. As a follow-up comment, how well could the insights from this study be generalized to other regions of the world?**

To address these points we have appended a new paragraph to the end of Sect. 5:

"We note here that the modifications listed in Table 1 are applied only to the nested grid simulations and not during model spin-up, which may introduce some inconsistencies with initial and boundary conditions. Because Mod. 1 addresses fundamental assumptions in the calculation of OC EFs, that particular modification may also apply to biomass burning in other regions of the world. Mods. 2 and 3, however, are specific to Southeast Asia in September of 2014, and other regions may similarly require further analysis in order to understand which uncertainties dominate locally and how they might impact our simulations through long-range transport."

**3. The impact of uncertainties in vertical transport was briefly mentioned in the paper. A recent study (Wizenberg et al., 2023) showed that the underestimation of model fire emissions relative to surface observations could be largely attributed to the injection height scheme in GEOS-Chem, which suggests transport could be a very important source of uncertainty. If another sensitivity run with changed injection height is not feasible, could the authors slightly expand the discussion on this in the paper?**

**Wizenberg, T., Strong, K., Jones, D. B. A., Lutsch, E., Mahieu, E., Franco, B., & Clarisse, L. (2023). Exceptional wildfire enhancements of PAN, C2H4, CH3OH, and HCOOH over the Canadian high Arctic during August 2017. Journal of Geophysical Research: Atmospheres, 128, e2022JD038052. https://doi.org/10.1029/2022JD038052**

Although we do not perform simulations specifically designed to test different injection schemes, we have considered insights on injection height from our comparison of biomass burning inventories (Fig. 4) and revised Section 4.1.3 as follows:

"A recent study showed that the underestimation of model fire tracers relative to surface observations could be largely attributed to the injection height scheme in GEOS-Chem (Wizenberg et al., 2023), which suggests that the vertical distribution of emissions could be an important source of model uncertainty. In recent versions of GEOS-Chem including 12.5.0, the default treatment of biomass burning emissions from GFED is to inject all emissions into the surface layer of the model. This is very likely not an accurate representation of all fires in Southeast Asia, but may be appropriate in particular for peat fires, which occur close to the ground and tend to produce plumes confined to altitudes below 1000 m (Tosca et al., 2011). Furthermore, partial injection of emissions above the surface layer would likely result in worse agreement with ground-based $PM_{2.5}$ observations (Fig. 3). Although we do not perform simulations specifically designed to test different injection schemes, our simulations with different biomass burning inventories (Fig. 4) provide some constraint on this issue. As with GFED, emissions from FINN are injected into the surface layer of the model, whereas emissions from GFAS and QFED are distributed across injection heights. Emissions from GFAS are injected evenly into each model layer between the surface and a mean altitude of maximum injection, while 65% of emissions from QFED are injected between the surface and the top of the planetary boundary layer (PBL) and the remaining 35% are injected between the top of the PBL and an altitude of 5500 m. As shown in Fig. 4, distributing emissions above the surface layer does not have a clear impact on model AOD across Southeast Asia in March, with QFED (NMB = -29%) performing somewhat better than GFED (NMB = -40%) and GFAS somewhat worse (NMB = -52%). Both inventories, however, perform notably worse (NMB <= -57%) than GFED (NMB = -36%) in September, when biomass burning is dominated by peat and might reasonably be described by emissions closer to the surface. We recognize that the vertical emission distribution is an important source of uncertainty in the model but do not necessarily recommend changing the injection scheme as implemented in GFED for simulations of Southeast Asia."

Tosca, M. G., Randerson, J. T., Zender, C. S., Nelson, D. L., Diner, D. J., and Logan, J. A.: Dynamics of fire plumes and smoke clouds associated with peat and deforestation fires in Indonesia, J. Geophys. Res. Atmos., 116, https://doi.org/10.1029/2010JD015148, 2011.

Wizenberg, T., Strong, K., Jones, D. B. A., Lutsch, E., Mahieu, E., Franco, B., and Clarisse, L.: Exceptional wildfire enhancements of PAN, $C_2H_4$, $CH_3OH$, and HCOOH over the Canadian high Arctic during August 2017, J. Geophys. Res. Atmos., 128, e2022JD038 052, https://doi.org/10.1029/2022JD038052, 2023.

**Other points:**

**Figure 1(b): It would be better to use scientific notation in the colorbar.**

We have changed the colorbar as suggested.

**Figure 3: Is there a reason why only SS1 is shown for March and only SS2 is shown for September?**

This is because sensitivity simulation SS2 contains modifications that only apply to the September case. We applied and ran each modification separately for September and the results are not shown in Fig. 3, but they are described in the text at line 317.

**L86: It would be great if the authors could clarify how the daily and 3-hourly scaling factors were applied.**

The daily and 3-hourly scaling factors are fractional values that are provided with the data and applied to the cumulative monthly dry matter emissions. We have updated line 86 to clarify this point: "...we configure GEOS-Chem to apply fractional daily and 3-hourly scaling factors that are provided with the cumulative monthly data so that we can achieve finer temporal resolution in the nested simulation."